# A 3D navigation template for guiding a unilateral lumbar pedicle screw with contralateral translaminar facet screw fixation: a study protocol for multicentre randomised controlled trials

Zhen-Xuan Shao,[1] Wei He,[2] Shao-Qi He,[3] Sheng-Lei Lin,[4] Zhe-Yu Huang,[5] Hong-Chao Tang,[6] Wen-Fei Ni,[1] Xiang-Yang Wang,[1] Ai-Min Wu[1]

► Prepublication history and additional material are available. To view these files please visit the journal online (http://dx.doi.org/10.1136/bmjopen-2017-016328).

## ABSTRACT

**Introduction** The incidence of lumbar disc degeneration disease has increased in recent years. Lumbar interbody fusion using two unilateral pedicle screws and a translaminar facet screw fixation has advantages of minimal invasiveness and lower costs compared with the traditional methods. Moreover, a method guided by a three-dimensional (3D) navigation template may help us improve the surgical accuracy and the success rate. This is the first randomised study using a 3D navigation template to guide a unilateral lumbar pedicle screw with contralateral translaminar facet screw fixation.

**Methods and analysis** Patients who meet the criteria of the surgery will be randomly divided into experimental groups and control groups by a computer-generated randomisation schedule. We will preoperatively design an individual 3D navigation template using CATIA software and MeditoolCreate. The following primary outcomes will be collected: screw angles compared with the optimal screw trajectories in 3D digital images, length of the wound incision, operative time, intraoperative blood loss and complications. The following secondary outcomes will be collected: visual analogue scale (VAS) for back pain, VAS for leg pain and the Oswestry Disability Index. These parameters will be evaluated on day 1 and then 3, 6, 12 and 24 months postoperatively.

**Ethics and dissemination** The study has been reviewed and approved by the institutional ethics review board of the Second Affiliated Hospital and Yuying Children's Hospital of Wenzhou Medical University. The results will be presented at scientific communities and peer-reviewed journals.

**Trial registration number** ChiCTR-IDR-17010466

### Strengths and limitations of this study

► This is the first multicentre randomised controlled trial to compare a three-dimensional (3D) navigation template for guiding a unilateral lumbar pedicle screw with contralateral translaminar facet screw fixation versus traditional multiple X-ray fluoroscopy.
► We will first use Mimics and CATIA software to design the 3D navigation template of a unilateral lumbar pedicle screw with contralateral translaminar facet screw fixation.
► The clinical data will be collected prospectively at least 24 months after the operation.
► Because this is a multicentre randomised controlled trial, the surgeons' experiments at different research sites may influence the outcomes.

screw fixation on both sides, which has gained popularity among general surgeons because of easiness, the lumbar interbody fusion, which uses two unilateral pedicle screws and a translaminar facet screw fixation, can reduce tissue trauma and the length of the wound incision, as well as promote recovery.[10–12]

The translaminar screw is a long screw that goes through the spinous process and zygapophyseal joints and ends in a lower vertebral facet to fix the other side of the joint.[13] The biomechanical properties of the translaminar screw are the same as pedicle screws in vitro biomechanical experiments.[14 15] Park *et al*[16] reported that the visual analogue scale (VAS) and Oswestry Disability Index (ODI) scores of the patient were significantly improved after minimally invasive surgery of anterior lumbar interbody fusion combined with the translaminar screw. Liu *et al*[11] reported a recovery rate of 93.5% through minimally invasive surgery of transforaminal lumbar interbody fusion combined with the translaminar screw.

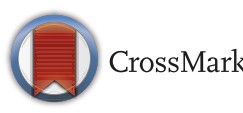

For numbered affiliations see end of article.

**Correspondence to**
Dr Ai-Min Wu; aiminwu2005@163.com

## INTRODUCTION

With the ageing of society, the incidence of lumbar disc degeneration disease is increasing,[1 2] which is a major socioeconomic burden.[3–5] If conservative treatment of lumbar degenerative disease is ineffective, surgical intervention is recommended for these patients.[6–9] Compared with the pedicle

**BMJ**

However, there is higher risk of nerve and vascular injury when the angle deviation of the translaminar screw or screw entry point is incorrect.[17] To achieve an optimal screw trajectory, many surgeons choose multiple X-ray images as a guide to locate the entrance point and adjust the angle, which will expose both patients and surgeons to considerable radiation.[11 12 18] The technique of three-dimensional (3D) reconstruction and a 3D rapid printed navigation template, which extracts the posterior surface features of the lumbar vertebrae, may help determine the optimal trajectory of translaminar facet screw fixation.

To the best of our knowledge, there are no randomised controlled studies of a 3D navigation template versus traditional multiple X-ray fluoroscopy for guiding a unilateral lumbar pedicle screw with contralateral translaminar facet screw fixation. In this study, we will conduct a randomised controlled trial (RCT) to compare 3D navigation template for guiding a unilateral lumbar pedicle screw with contralateral translaminar facet screw fixation versus traditional multiple X-ray fluoroscopy.

## METHODS AND ANALYSIS

The study has been reviewed and approved by the institutional ethics review board of the Second Affiliated Hospital and Yuying Children's Hospital of Wenzhou Medical University. All of the participants will sign their informed consent. The protocol has been registered in the Chinese Clinical Trial Registry (ChiCTR), assigned to be the representative registry of China to join the WHO ICTRP in 2007, with protocol number ChiCTR-IDR-17010466. The 2013 SPIRIT checklist (see online supplementary table 1) was used to check our reports.[19 20]

### Participants

This study is a parallel group RCT conducted at the Orthopaedic Hospital, Second Affiliated Hospital and Yuying Children's Hospital of Wenzhou Medical University.

### Inclusion criteria

1. Adult patients older than 18 years;
2. Lower back pain that is chronic or combined with neurological symptoms of the lower extremities;
3. Single-segment lower lumbar vertebral disease (including lumbar spinal canal stenosis, foraminal stenosis, segmental instability, lumbar disc herniation and painful disc degeneration (back disc)); and
4. Inefficacy after strict conservative treatment for more than 6 months.

### Exclusion criteria

1. Serious deformity of the lumbar vertebrae;
2. Dysplasia of the lumbar pedicle or vertebral lamina;
3. Obvious osteoporosis of the lumbar vertebrae;
4. Metabolic bone diseases such as osteomalacia or Paget's disease;
5. Spondylolisthesis grade >2 (Meyerding);
6. Cauda equina injury or severe radiculopathy;
7. Postinflammatory instability of the vertebral spine;
8. Body mass index >30;
9. Immunological diseases or metabolic syndrome;
10. Therapy with systematic corticosteroids or immunosuppressants; and
11. Current use of Coumadin (warfarin) or heparin therapy for more than 6 months at the time of the operation.

### Sample size calculation

We performed a power analysis to assess the required sample size to show safety with a power (1–b) of 0.8 and α of 0.05. Based on the pre-experiment and related literature,[21] the proportion of the control group was set to 84.6%, and the proportion of intervention group was set to 95%. We performed a two independent proportions power analysis on PASS (Power Analysis and Sample Size), and the result was 132.

### Randomisation and blinding

Patients will be randomly divided into an experimental group and a control group using a computer-generated randomisation schedule. The random sequence will be stored in a sealed opaque envelope to ensure that the patients, personnel and the outcome assessor will be blinded to the group allocation. Until the last questionnaires have been completed, the participant's allocated intervention will be revealed by the study secretary. If a patient's condition deteriorates, the blinding experiment will be terminated, and the patient's safety will be given the highest priority (figure 1).

### Monitoring

After screening, patients meeting the inclusion criteria should participate in the study as long as possible. If not, we must save the record to judge bias. Meanwhile, recording the patient's basic information is necessary to discuss the results of the experiment.

Before the experiment, we will explain the purpose of the experiment and the importance of the experimental procedures, to obtain consent and ideal compliance. If the selected patient is unqualified, elimination will be undesirable, and the data of the patient will be substituted by the average of the group. Furthermore, loss to follow-up is inevitable; the data of these patients will also be replaced by the average.

Surgeons and data collectors will undergo training, and a preliminary experiment involving the data measurement and questionnaire will be conducted in advance. Recording the time and staff is necessary when the collector gathers the data, which will be convenient to review.

### Study procedure

A retrospective review of all patients will be conducted by examining their medical records before the operation. The involved segments were identified via a C-arm machine and then marked on the patient's skin. The patients' CT images with a thickness less than 1 mm will be exported from the Star PACS image system of the

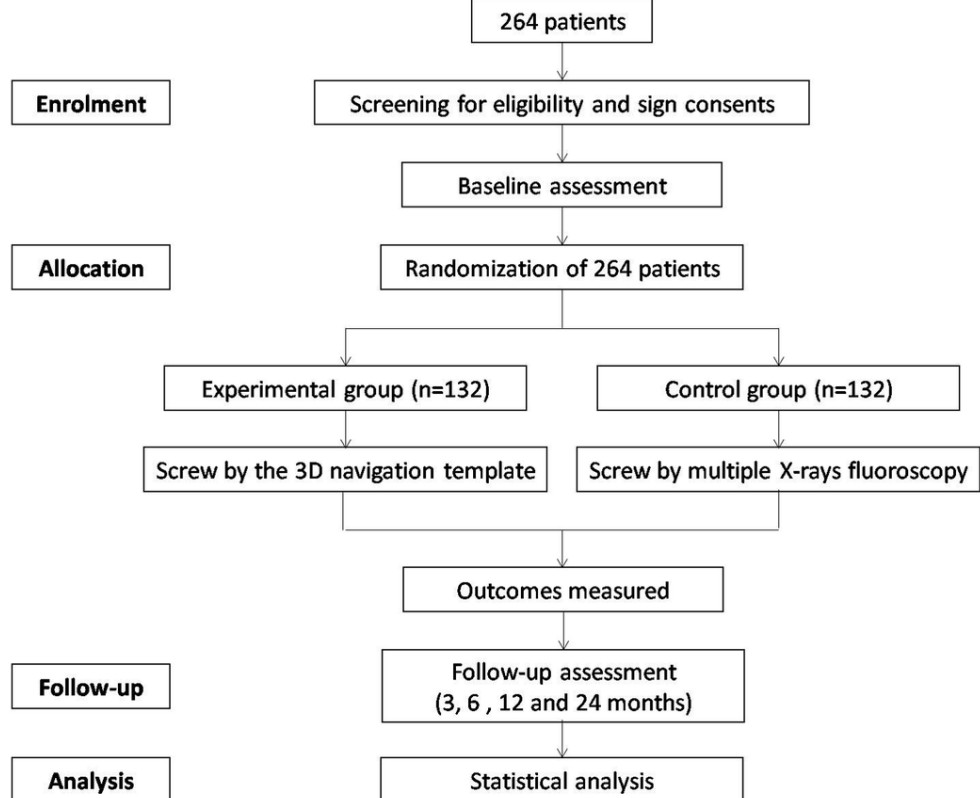

**Figure 1** The flow chart of randomised controlled trial.

hospital and imported into the Mimics 16.0 to construct 3D images, in which we adjust the threshold depending on the bone density. We regard the two adjacent segments as a functional segmental unit (FSU), such as L2-3, L3-4, L4-5 or L5-S1.

Then, the individual 3D navigation templates will be designed and printed using CATIA software and MeditoolCreate. The detailed steps are as follows: the FSU digital data in the Mimics 16.0 will be saved in STL format and imported into CATIA software. We will establish the optimal screw trajectory, which includes two vertebral pedicle screws trajectories and one contralateral translaminar facet screw trajectory. We will measure three angles of the translaminar facet screw trajectory: $\alpha$, $\beta$ and $\gamma$; $\alpha$ is the angle between the screw trajectory and the vertical line in the anteroposterior view; $\beta$ is the angle between the screw trajectory and the upper endplate of the lower vertebra in the lateral view; $\gamma$ is the angle between the screw trajectory and the midline of the lumbar vertebra in the top view (figure 2). The pedicle screw trajectories will be designed according to method of Roy-Camille.[22] We will measure two angles, $\delta$ and $\varepsilon$, which is the angle between the ideal screw trajectory and the midline of the lumbar vertebra in the top view, respectively (figure 2).

After the optimal screw trajectories were determined, the curved posterior surface of the lumbar vertebra around the screw entry point was extracted to create the 3D navigation template (figure 3). Then, we imported the 3D navigation template file into the 3D printer to rapid print it through MeditoolCreate (figure 4). The experiment on

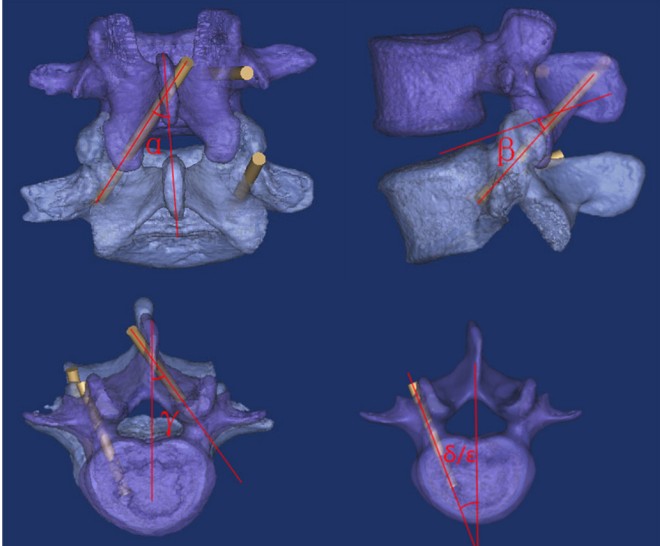

**Figure 2** The designed pedicle screw trajectory and translaminar facet screw trajectory: $\alpha$ is the angle between the translaminar facet screw trajectory and the vertical line in the anteroposterior view; $\beta$ is the angle between translaminar facet screw trajectory and the upper endplate of lower vertebra in the lateral view; $\gamma$ is the angle between translaminar facet screw trajectory and midline of the lumbar vertebra in the top view; $\delta$ and $\varepsilon$ are the angles between pedicle screw trajectory (upper vertebra and lower vertebra) and midline of the lumbar vertebra in the top view.

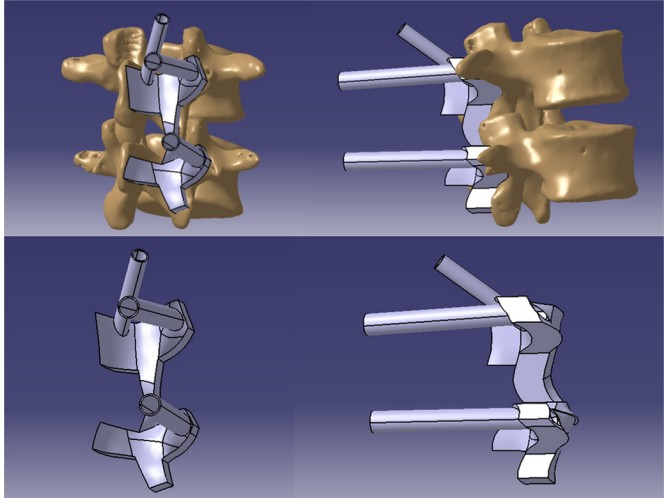

**Figure 3** The designed 3D navigation template in CATIA software.

3D printed lumbar model will be conducted before clinical to make sure the accurate screw trajectory (figure 5).

The 3D navigation template will be sterilised and used to guide the screw in further operations on the patients. In experimental group, the dissection of the paraspinal tissue will performed at one side to expose the part of spinous process, laminar and facet joint; the 3D navigation template will be attached to the surface of the above bony construction. Then, the Kirschner wire (K-wire) will be inserted inside the 3D navigation template, and a trajectory for the screw will be fashioned with a cannulated drill; screws will be subsequently introduced. In the control group, multiple X-ray fluoroscopy will be used, and the screw trajectory will be adjusted by intraoperative X-ray fluoroscopy.

After the operation, CT scans of the patients in both groups will be obtained, and the $\alpha 1$, $\beta 1$, $\gamma 1$, $\delta 1$ and $\epsilon 1$ angles will be measured corresponding to the angles of the optimal screw trajectory in 3D images. The placement and position of the screws will be divided into three types (type I, the screw does not penetrate out of the cortex; type II, the screw partially penetrates out of the cortex, but by less than 2 mm; and type III, the screw penetrates out of the cortex by more than 2 mm).

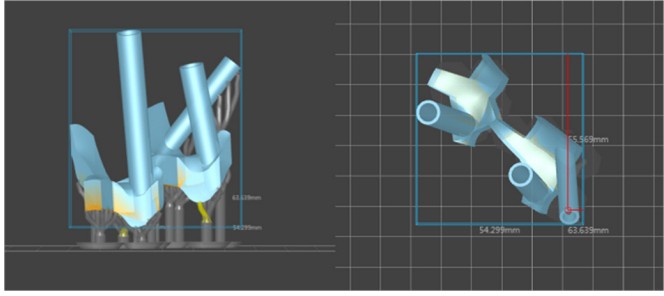

**Figure 4** 3D navigation template imported into MeditoolCreate software to print.

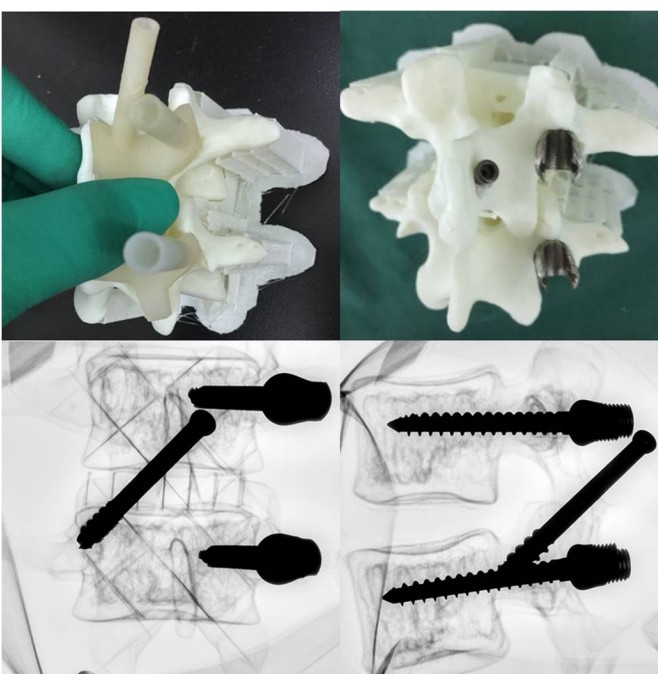

**Figure 5** One example of the experiment on conducted 3D printed lumbar model before clinical application to make sure the accurate screw trajectory.

## Outcome measure

### Primary outcomes (table 1)
1. Screw angles compared with the optimal screw trajectories in the 3D digital images and the position of the screws.
2. Wound incision length, operative time, intraoperative blood loss, times of X-ray exposure and complications.

### Secondary outcomes (table 1)
1. VAS for back pain;
2. VAS for leg pain; and
3. ODI.

The data will be collected preoperatively; on day 1 postoperatively; and then at 3, 6, 12 and 24 months postoperatively.

### Statistical analysis
All data will be analysed using SPSS V.19.0 software. The differences in wound incision length, operative time, intraoperative blood loss and angles between the two groups will be analysed using a two independent-sample t-test with an $\alpha$ of 0.05. VAS and ODI scores at preoperation, postoperation, 3 months postoperatively, 6 months postoperatively and 12 months postoperatively will be analysed using a repeated-measures analysis of variance. The times of X-ray exposure and frequency of screw penetration out of the cortex will be compared using the $\chi^2$ test.

## DISCUSSION
Translaminar facet screw fixation has been reported as a minimally invasive technique,[11 12 23] and biomechanical

**Table 1** The data need to collect through the research

| Assessment | Baseline −1 day | Perioperation Day 0 | Day 1 | 3 months | 6 months | 12 months | 24 months |
|---|---|---|---|---|---|---|---|
| | | | | Postoperation | | | |
| Eligibility criteria | × | | | | | | |
| Recruitment | × | | | | | | |
| Informed consent | × | | | | | | |
| Randomisation | × | | | | | | |
| Baseline demographics | × | | | | | | |
| Medical history | × | | | | | | |
| X-ray exposure | | × | | | | | |
| Wound incision length | | × | | | | | |
| Operative time | | × | | | | | |
| Blood loss | | × | | | | | |
| Complications | | × | × | × | × | × | × |
| Screw angles | | | | | | | |
| α1 | | | × | | | | |
| β1 | | | × | | | | |
| γ1 | | | × | | | | |
| δ1 | | | × | | | | |
| ε1 | | | × | | | | |
| The position of the screws | | | × | | | | |
| VAS of back pain | × | | × | × | × | × | × |
| VAS of leg pain | × | | × | × | × | × | × |
| ODI | × | | × | × | × | × | × |

ODI, Oswestry Disability Index; VAS, visual analogue scale.

comparisons between facet screw and pedicle screw fixation have revealed biomechanical equivalence between them.[14 15 24] Unilateral lumbar pedicle screw may be used by some surgeons; however, Sethi et al[25] used in vitro human cadaveric lumbar spines to compare the biomechanical properties of bilateral pedicle screws, unilateral pedicle screws (UPSs), UPSs and translaminar facet screws, and unilateral single pedicle screws and translaminar facet screws (V construct). They found that the UPS construct was the least stable in all loading modes and that the added translaminar facet screw enhanced the stability and stiffness. Similar results were obtained from a finite element analysis study.[26] The biomechanical study of Luo et al[27] revealed that a significantly larger displacement of the contralateral articular process was recorded in a model with only unilateral pedicle screws, and they reported that the unilateral pedicle screw combined with a contralateral translaminar facet screw had an instant and long-term equivalent biomechanical ability to that of

the traditional bilateral pedicle screw (making it an alternative to the bilateral pedicle screw) and could be less invasive while maintaining stable and effective instrumentation.

Current computer-assisted surgery systems can achieve the accurate screw trajectory; however, the cost of computer-assisted surgery systems is considerable, and the manipulation is too complicated,[28] whereas the accuracy remains dependent on multiple radiations.

Combining 3D reconstruction and 3D rapid printing is a new mode of production, which is applied in many fields because of the advantages of visualisation, plasticity and rapid printing.[29–31] Moreover, 3D printing templates can help surgeons to achieve accurate screw placement for posterior cervical fixation and scoliosis.[32 33]

In this study, we will compare the 3D navigation template to multiple X-ray fluoroscopy for guiding a unilateral lumbar pedicle screw with contralateral translaminar

 

facet screw fixation. The compared results will include the perioperative parameters (such as X-ray exposure, wound incision length, operative time, blood loss); screw angles of $\alpha 1$, $\beta 1$, $\gamma 1$, $\delta 1$ and $\epsilon 1$; postoperative complications; the VAS of back and leg pain; and the ODI.

The main strength of the study is the design as an RCT rather than an observational comparative study. An RCT has the advantage of controlling all possible variables due to the random sequence generation as opposed to observational studies, where confounding and bias may be more problematic. High-quality RCTs are generally regarded as the gold standard for studying the effectiveness of an intervention.

Because of some patients' death, relocation or other reasons, the representativeness of the sample is destroyed. Meanwhile, if some patients accept experimental treatment measures such as therapeutic drugs, the results will appear to be deviated. For these experiment biases, strict monitoring can reduce the loss of follow-up and can also improve patient compliance, through which bias will decrease.

## ETHICS AND DISSEMINATION

The study had been reviewed and approved by the institutional ethics review board of the Second Affiliated Hospital and Yuying Children's Hospital of Wenzhou Medical University. The procedure will be performed following the principles described in the Declaration of Helsinki. All of the participants will sign their informed consent. The protocol has been registered in the ChiCTR, assigned to be the representative registry of China to join the WHO ICTRP in 2007, with protocol number ChiC-TR-IDR-17010466.

We will share individual patient data within 2 years after the trial is completed, and the original data will be collected using a clinical recording formula (both paper and electronic versions). The results will be presented at scientific communities (such as the International Congress of the Chinese Orthopaedic Association) and peer-reviewed journals.

**Author affiliations**
[1]Department of Spine Surgery, Orthopaedic Hospital, The Second Affiliated Hospital and Yuying Children's Hospital of the Wenzhou Medical University, The Second Medical School of the Wenzhou Medical University, Zhejiang Spine Center, Wenzhou, China
[2]Department of Orthopaedics, People's Hospital of Shaoxing, Zhejiang University Shaoxing Hospital, Shaoxing, China
[3]Department of Orthopaedics, People's Hospital of Ruian, The Third Affiliated Hospital of Wenzhou Medical University, Ruian, China
[4]Department of Orthopaedics, Wenzhou Center Hospital, Dingli Hospital of Wenzhou Medical University, Wenzhou, China
[5]Department of Orthopaedics, Ningbo No. 6 Hospital, Ningbo, Zhejiang, China
[6]Department of Orthopaedics, Jiaxing Hospital of Zhejiang General Corps of Armed Police Forces, Jiaxing, China

**Contributors** XYW and AMW are the principal investigators of this trial. ZXS, WFN, XYW and AMW contributed to the development of the research question and trial design. ZXS and AMW contributed to the drafting of the manuscript. ZXS, WH, SQH, SLL, ZYH and HCT are responsible for the recruitment and treatment of patients. All authors edited the final manuscript. All authors read and approved the final manuscript.

**Funding** This work was funded by the National Natural Science Foundation of China (81501933), Zhejiang science and technology innovation project (2016R413061), Wenzhou Science andTechnology Project (Y20160369).The funders had no role in the design, execution, or writing of the study.

**Competing interests** None declared.

**Ethics approval** The study had been reviewed and approved by the institutional ethics review board of the Second Affiliated Hospital and Yuying Children's Hospital of the Wenzhou Medical University. The protocol has been registered in Chinese Clinical Trial Registry (ChiCTR), assigned to be the representative registry of China to join WHO ICTRP in 2007, with the protocol number ChiCTR-IDR-17010466.

**Provenance and peer review** Not commissioned; externally peer reviewed.

**Data sharing statement** The full data set will be made available when this trial is completed and published. Application for the data to be released should be made in contact to AMW (principle investigator).

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
