## [Reviewer comments · BMJ Open]

ARTICLE DETAILS

TITLE (PROVISIONAL)	A 3D navigation template for guiding a unilateral lumbar pedicle screw with contralateral translaminar facet screw fixation: A study protocol for multicentre randomized controlled trials.
AUTHORS	Shao, Zhen-Xuan; He, Wei; He, Shao-Qi; Lin, Sheng-Lei; Huang, Zhe-Yu; Tang, Hong-Chao; ni, wenfei; Wang, Xiang-Yang; Wu, Ai-Min

VERSION 1 - REVIEW

REVIEWER	Yung Park National Health Insurance Service Ilsan Hospital Republic of Korea
REVIEW RETURNED	21-Feb-2017

GENERAL COMMENTS	This protocol has a interesting topic in the field of spinal surgery. I believe this article is ready for publication.
--

REVIEWER	Roger Hartl Weill Cornell Brain and Spine USA
REVIEW RETURNED	02-Mar-2017

GENERAL COMMENTS	The authors report a multi-center trial using a 3D template to guide ipsilateral pedicle screws and contralateral trans-laminar screws for posterior fixation. This is an interesting and real worthwhile endeavor as it allows individualized instruments to be used for patient care, a novel concept. The authors describe the appropriate collection of primary and secondary endpoints and have well illustrated figures. Unfortunately, the wording is extremely poor, which does not help get the authors message across to exactly what they are trying to achieve with this paper. The figures are good; however, the tables do not go into detail about any of the outcome measures that the authors intended to study. The results being presented in a different paper and "international spine related meetings" is not appropriate. This is an academic paper and should follow the well established guidelines for presenting information to the scientific community. The discussion is poorly written and speaks in the "future tense." The paper needs to be written to convey information and results, not a plan.
---

REVIEWER	Yufu Wang Harbin Medical University
REVIEW RETURNED	23-Apr-2017

GENERAL COMMENTS	This study reported a randomized controlled trials data concerning the “The 3D navigation template in guiding unilateral lumbar pedicle screw with 2 contralateral translaminar facet screw fixation” .This is the first multicenter randomized controlled trials to compare 3D navigation 10 template in guiding unilateral lumbar pedicle screw with contralateral translaminar.However, some issues shou be addressed. 1.Unilateral lumbar pedicle screw is enough to provide stablisation. Is there any benefits of combining with contralateral translaminar facet screw fixation? It should be discussed. 2.No post-surgery images or cases were provided to illustrate the advantages of this protocol. 3. Language should be revised.
---

VERSION 1 – AUTHOR RESPONSE

Reviewer: 1

Yung Park

National Health Insurance Service Ilsan Hospital, Republic of Korea

Please state any competing interests or state ‘None declared’: None

Please leave your comments for the authors below

This protocol has a interesting topic in the field of spinal surgery.

I believe this article is ready for publication.

Answer: Great appreciate for you careful review and positive comments.

Reviewer: 2

Roger Hartl

Weill Cornell Brain and Spine, USA

Please state any competing interests or state ‘None declared’: None Declared

Please leave your comments for the authors below

Comment 01: The authors report a multi-center trial using a 3D template to guide ipsilateral pedicle screws and contralateral trans-laminar screws for posterior fixation. This is an interesting and real worthwhile endeavor as it allows individualized instruments to be used for patient care, a novel concept. The authors describe the appropriate collection of primary and secondary endpoints and have well illustrated figures.

Answer: Great appreciate for you careful review and positive comments.

Comment 02: Unfortunately, the wording is extremely poor, which does not help get the authors message across to exactly what they are trying to achieve with this paper. The figures are good; however, the tables do not go into detail about any of the outcome measures that the authors intended to study. The results being presented in a differer paper and “international spine related meetings” is not appropriate. This is an academic paper and should follow the well established guidelines for presenting information to the scientific community. The discussion is poorly written and speaks in the “future tense.” The paper needs to be written to convey information and results, not a plan.

Answer: Great thanks for your careful review and point out of this. As your suggestion, we let the manuscript edited by “AJE (<https://www.aje.com/>)”, we hope our manuscript can be readily understand by readers now. Then, we add the angle of “ α_1 , β_1 , γ_1 δ_1 and ϵ_1 ” that we intended to measure into the Table 1. And we change “the results being presented in a different paper and “international spine related meetings”” to “The results will be presented at scientific communities (such

as International Congress of Chinese Orthopaedic Association) and peer-reviewed journals.(Page 15 Line 22 -Page 16 Line 2)". Moreover, we convey information and results in revised manuscript, and add "In this study, we will compare the 3D navigation template to multiple X-rays fluoroscopy in guiding unilateral lumbar pedicle screw with contralateral translaminar facet screw fixation. The compared results will be include the peri-operative parameters (such as X-ray exposure, wound incision length, operative time, blood loss) , screw angles of α_1 , β_1 , γ_1 , δ_1 , and ϵ_1 , post-operative complications and VAS of back and leg pain, and ODI." In discussion (Page 14 Line 17-22). Thank you again for your careful review and valuable comments.

Reviewer: 3

Yufu Wang

Harbin Medical University

Please state any competing interests or state 'None declared': None

Please leave your comments for the authors below

Comment 01: This study reported a randomized controlled trials data concerning the "The 3D navigation template in guiding unilateral lumbar pedicle screw with 2 contralateral translaminar facet screw fixation" .This is the first multicenter randomized controlled trials to compare 3D navigation template in guiding unilateral lumbar pedicle screw with contralateral translaminar.However, some issues should be addressed.

Answer: Great appreciate for you careful review and positive comments. And the issue you point out will be responded point by point as following.

Comment 02 : Unilateral lumbar pedicle screw is enough to provide stabilisation. Is there any benefits of combining with contralateral translaminar facet screw fixation? It should be discussed.

Answer: Great appreciate for you careful review and point out of this. It's true that the unilateral lumbar pedicle screw may used by some surgeons, however, Sethi et al¹ used vitro human cadaveric lumbar spines to compare the biomechanical properties of bilateral pedicle screws (BPS), unilateral pedicle screws (UPS), UPS and translaminar facet screw (UPS+TLFS); and unilateral single pedicle screw and translaminar facet screw (V construct), they found that unilateral pedicle screws (UPS) construct was the least stable in all loading modes, the added translaminar facet screw will enhance the stability and stiff. Similar results showed by finite element analysis study². The biomechanical study of Luo et al³ found that a significantly larger displacement of contralateral articular process was recorded in unilateral pedicle screws only model, and recommended the unilateral pedicle screw combined with contralateral translaminar facet screw had instant and long-term equivalent biomechanical ability to that of traditional bilateral pedicle screw, making it an alternative option to bilateral pedicle screw, and could be less invasive while maintains a stable and effective instrumentation. Therefore, we use translaminar facet screw in this study. As your suggestion here, we add "Unilateral lumbar pedicle screw may be used by some surgeons, however, Sethi et al¹ used in vitro human cadaveric lumbar spines to compare the biomechanical properties of bilateral pedicle screws (BPSs), unilateral pedicle screws (UPSs), UPSs and translaminar facet screws (UPSs+TLFSs), and unilateral single pedicle screws and translaminar facet screws (V construct). They found that the unilateral pedicle screw (UPS) construct was the least stable in all loading modes and that the added translaminar facet screw enhanced the stability and stiffness. Similar results were obtained from a finite element analysis study². The biomechanical study of Luo et al³ revealed that a significantly larger displacement of the contralateral articular process was recorded in a model with only unilateral pedicle screws, and they reported that the unilateral pedicle screw combined with a contralateral translaminar facet screw had an instant and long-term equivalent biomechanical ability to that of the traditional bilateral pedicle screw (making it an alternative to the bilateral pedicle screw) and could be less invasive while maintaining stable and effective instrumentation." in revised manuscript (Page 13 Line 16 – Page 14 Line 8). Thank you again for your valuable comment.

Comment 03. No post-surgery images or cases were provided to illustrate the advantages of this protocol.

Answer: Great thanks for your careful review and point out of this. The study is still at the stage of the preparation and training to consistent the researchers from different centers now, therefore, without the post-surgery image of patients, however, we have the images that validation the 3D navigation template on 3D printed lumbar model. As your suggestion here, we add "Figure 5: One example of the experiment on conducted 3D printed lumbar model before clinical application to make sure the accurate screw trajectory" in revised manuscript. Thanks again.

Comment 04. Language should be revised.

Answer: Great thanks for your careful review and point out of this. As your suggestion, we let the manuscript edited by "AJE (<https://www.aje.com/>)", we hope our manuscript can be readily understand by readers now.

Finally, we express our kindest regards and greatest appreciate for your careful review, volunteer time, and so many valuable and constructive suggestions. Because all of them, our manuscript was significantly improved.

Reference

1. Sethi A, Muzumdar AM, Ingalhalikar A, et al. Biomechanical analysis of a novel posterior construct in a transforaminal lumbar interbody fusion model an in vitro study. *Spine J* 2011;11(9):863-9.
2. Gong Z, Chen Z, Feng Z, et al. Finite element analysis of 3 posterior fixation techniques in the lumbar spine. *Orthopedics* 2014;37(5):e441-8.
3. Luo B, Yan M, Huang J, et al. Biomechanical study of unilateral pedicle screw combined with contralateral translaminal facet screw in transforaminal lumbar interbody fusion. *Clin Biomech (Bristol, Avon)* 2015;30(7):657-61.